# Pharmacogenomic Profile and Adverse Drug Reactions in a Prospective Therapeutic Cohort of Chagas Disease Patients Treated with Benznidazole

**DOI:** 10.3390/ijms22041960

**Published:** 2021-02-16

**Authors:** Lucas A. M. Franco, Carlos H. V. Moreira, Lewis F. Buss, Lea C. Oliveira, Roberta C. R. Martins, Erika R. Manuli, José A. L. Lindoso, Michael P. Busch, Alexandre C. Pereira, Ester C. Sabino

**Affiliations:** 1Department of Infectious Disease and Institute of Tropical Medicine (IMT-SP), University of São Paulo, Av. Dr. Enéas Carvalho de Aguiar, 470, São Paulo 05403-000, Brazil; lewisbuss@gmail.com (L.F.B.); lea.c.oliveira@gmail.com (L.C.O.); betacristina@gmail.com (R.C.R.M.); erikamanuli@gmail.com (E.R.M.); sabinoec@gmail.com (E.C.S.); 2Institute of Infectology Emílio Ribas, São Paulo 01246-900, Brazil; jlindoso@usp.br; 3Blood Systems Research Institute, San Francisco, CA 94118, USA; mbusch@vitalant.org; 4Department of Laboratory Medicine, University of California San Francisco, San Francisco, CA 94143, USA; 5Department of Genetics, Harvard Medical School, Boston, MA 02115, USA; alexandre.pereira@incor.usp.br; 6Laboratory of Genetics and Molecular Cardiology, The Heart Institute, University of São Paulo, São Paulo 05403-000, Brazil

**Keywords:** Chagas disease, benznidazole, adverse drug reactions, pharmacogenomics

## Abstract

Chagas disease remains a major social and public health problem in Latin America. Benznidazole (BZN) is the main drug with activity against *Trypanosoma cruzi*. Due to the high number of adverse drug reactions (ADRs), BZN is underprescribed. The goal of this study was to evaluate the genetic and transcriptional basis of BZN adverse reactions. Methods: A prospective cohort with 102 Chagas disease patients who underwent BZN treatment was established to identify ADRs and understand their genetic basis. The patients were classified into two groups: those with at least one ADR (*n* = 73), and those without ADRs (*n* = 29). Genomic analyses were performed comparing single nucleotide polymorphisms between groups. Transcriptome data were obtained comparing groups before and after treatment, and signaling pathways related to the main ADRs were evaluated. Results: A total of 73 subjects (71.5%) experienced ADRs. Dermatological symptoms were most frequent (45.1%). One region of chromosome 16, at the gene LOC102724084 (rs1518601, rs11861761, and rs34091595), was associated with ADRs (*p* = 5.652 × 10^−8^). Transcriptomic data revealed three significantly enriched signaling pathways related to BZN ADRs. Conclusions: These data suggest that part of adverse BZN reactions might be genetically determined and may facilitate patient risk stratification prior to starting BZN treatment.

## 1. Introduction

The protozoan parasite *Trypanosoma cruzi* is the causal agent of Chagas disease (CD), a neglected tropical disease. Although the global prevalence of CD has fallen from 18 million in 1991 to 5.7 million in 2010 due to effective vector control programs [1,2,3], it still represents a major concern in endemic settings. Most people infected with *T. cruzi* live in Mexico, Central America, and South America [3]. For example, in Brazil, one million people are currently estimated to be infected with the parasite. It is notable then, given the substantial disease burden, that therapeutic options are limited, and benznidazole (BZN) is the only drug available for treatment in Brazil [4].

BZN is poorly tolerated due to frequent adverse drug reactions (ADRs), with 20–25% of adult patients unable to complete the standard 60-day treatment course [5,6,7,8]. Different BZD schemes have been proposed by observational studies, including shorter courses and even intermittent treatment. These may lead to comparable clinical and serological outcomes and, especially, to fewer ADRs and better compliance [9,10].

Despite this major limitation, few studies have explored the biological basis for increased ADR susceptibility. It was previously shown that the development of BZN-related ADRs is independent of dose [11]. In addition, BZN ADRs are negatively associated with Black race and positively associated with female sex and schooling [12]. Notably, however, only one study investigated the genetic basis of BNZ-associated ADRs, in which patients carrying HLA-B*3505 were more likely to abandon treatment [13].

Therefore, a better understanding of the pharmacogenetic aspects of BZN could significantly impact the management of patients. The purpose of this study was to conduct a multi-omics evaluation of BZN ADRs and describe genetic and transcriptional features associated with ADRs among individuals undergoing BZN treatment in a therapeutic cohort of patients chronically infected with *T. cruzi*.

## 2. Results

A total of 102 subjects were enrolled to receive BZN. Of these, 73 (71.5%) experienced at least one ADR and 22 (21.6%) did not complete the full treatment scheme due to severe ADRs. Table 1 summarizes their clinical and epidemiological characteristics. None of the variables evaluated were associated with ADRs, except for the Chagas disease form, with patients presenting with the cardiac form having a lower rate of ADRs.

Of the 102 patients, dermatological, gastrointestinal, and neurological ADRs were reported by 46 (45.1%), 30 (29.4%), and 28 (27.4%) patients, respectively. Treatment interruption was more frequent among patients who reported an ADR during the first 15 days of treatment with relative risk (RR) of 14.1, 95% confidence interval (CI) 2.0–100. Dermatological ADRs were strongly associated with treatment cessation compared to individuals who did not develop this type of ADR (RR 5.1, 95% CI 2.2–11.9). Other less frequent ADRs included arthralgia (12 subjects, 11.8%), reduced renal function (5 subjects, 4.9%), myelotoxicity (4 subjects, 3.9%), and hepatotoxicity (4 subjects, 3.9%). No ADRs required hospital admission, and most were controlled with a short course of low-dose corticosteroids, antihistamines, and analgesics. Sixteen patients (15.6%) reported ADRs on all study visits during treatment. Figure 1 summarizes the frequency of ADRs over time. Dermatological ADRs were more common at the beginning of treatment.

### 2.1. Genome-Wide Association Study Results

Of the initial 96 genotyped samples, 6 were excluded because they showed high kinship with other samples. A total of 90 patients were included in the Genome-Wide Association study (GWAS) analysis: 61 with at least one ADR (group gADR) and 29 without any ADRs (group gNADR). No single nucleotide polymorphism (SNP) reached our predefined genome-wide significance threshold. Nonetheless, a number of markers showed a strong association with ADR occurrence. Table 2 presents the 30 SNPs with the lowest *p*-values.

On chromosome 16, rs1518601, rs11861761, and rs34091595 SNPs (*p*-value = 5.652 × 10^−8^) were associated with the occurrence of ADRs. Figure 2 describes the region in detail, with many adjacent SNPs in linkage disequilibrium, categorized by *r*^2^ values between 0.6 and 0.8. These SNPs are located at the gene *LOC102724084*.

On chromosome 3, the SNP rs3968927 (*p*-value = 2.945 × 10^−7^) was in strong linkage disequilibrium with other six SNPs, with a *p*-value below 1.95 × 10^−6^, and several other dispersed across the genes LMOD3, ARL6IP5, FRMD4B, UBA3, TMF1, MIR3136, EOGT, and FAM19A4. On chromosome 15, there were five highly associated SNPs in strong linkage disequilibrium (*p*-value = 6.9 × 10^−7^) and in proximity to the genes GABRG3 and OCA2. However, none of these genes have a clear biological relationship with ADRs, drug metabolism, CD, or direct interactions with the immune system or inflammatory response. For a better representation of GWAS results and to assess the quality of the data, a Manhatan plot and a Quantile-quantile plot are provided in the Appendix A.

### 2.2. Transcriptomic Results

To understand the impact of BZN on gene expression, we compared samples prior to and at the end of BZN treatment (T0d × T60d). A total of 39 patients, 26 with at least one ADR (gADR) and 13 without any ADR (gNADR), were included in this analysis. There were 11 differentially expressed genes (DEGs) in the gNADR and 173 in the gADR group (comparing before and after BZD use). There were no DEGs in common (Appendix A). The gNADR group presented two activated signaling pathways that were completely independent of the 25 pathways in the gADR group (Appendix A). The Ingenuity Pathway Analysis (IPA) score value classification identified three main pathways in the gADR group, different from those found in the gNADR group.

Patients with dermatological ADRs had the most significantly enriched pathway (28 upregulated genes of 34 genes comprising the pathway, 82%). This was clearly associated with the biology of the skin (Figure 3).

Gastrointestinal ADRs, the second most common in this cohort, were related to a pathway linked with gastrointestinal functioning and disease. This pathway had a score of 39 and consisted of 34 genes, 79.4% of which were upregulated, indicating a strong relationship with gastrointestinal clinical manifestations developed in patients with ADRs (Figure 4).

Neurological manifestations were associated with a short pathway consisting of two genes. However, the GABRB1 gene had a log Fold change (logFC) above three, the highest gene upregulation observed, and a 200% increase compared to the pretreatment period (Figure 5).

## 3. Discussion

To the best of our knowledge, this is the first paper to describe genetics and transcriptomics data in relation to BZN-induced ADRs in CD patients. In our study, most patients (71.6%) experienced ADRs, in line with previous findings [14,15,16]. Dermatological events were the most frequent ADR observed, representing half of the total ADRs. This value is higher than that previously reported [15], likely due to differences in the clinical assessment procedures. One in five patients did not complete treatment due to ADRs. Dermatological events were most strongly associated with incomplete treatment (relative risk of 5.1 compared to those without this type of ADR) and tended to occur in the first two weeks of treatment. Aldasoro et al. (2018) reported that 80% of BZN-related ADRs occur within the first 30 days of treatment. Complaints such as arthralgia and neurotoxicity were more frequent during the last weeks of the standard 60-day treatment regimen, again in agreement with other studies [14,15,16]. We did not find any difference regarding demographic characteristics and ADR frequency, in disagreement with what was reported in other studies [12], which found a higher frequency of ADRs and BZD discontinuation in women and in those who graduated from elementary school. Maybe our study population represents a cosmopolitan region where the population’s demographic heterogeneity is less evident.

Although the sample size was small for a GWAS, we detected three SNPs (rs1518601, rs11861761, and rs34091595) that were highly associated with the occurrence of ADRs. The SNPs are intronic variants located at the gene LOC102724084 on chromosome 16. Unfortunately, the function of this gene is unknown, so we cannot verify the biological plausibility of this finding. There were several other adjacent SNPs in strong linkage disequilibrium located in the same region, many with *p*-values less than 10^−5^, suggesting that this is a true finding. There is only one previous study conducted in Bolivia that attempted to detect genes associated with BZN ADRs [13]. In that cohort, the gene HLA-B 3505 was found to be associated with adverse reactions. However, this human leukocyte antigen (HLA) is not frequent in the Brazilian population, and we did not confirm their findings.

We identified substantially different gene expression profiles between patients with and without adverse reactions. In the ADR group, there were more differentially expressed genes at the end of treatment compared to pretreatment. Dermatological manifestations were correlated with a pathway in which 82.4% of the genes were upregulated. These genes are associated with many dermatological conditions. They have previously been associated with more severe clinical expression of dermatitis [17,18,19,20] and advanced stages of atopic dermatitis [21], increased epidermal hyperplasia [22,23], cyclic itching associated with epidermolytic hyperkeratosis [24], lichen planus [25,26], erythema nodosum [27], and ichthyosis with confetti [28]. Many of the dermatological ADRs reported by patients can be directly associated with this pathway. The pathway linked to gastrointestinal manifestations showed significant upregulation of 79.4% of the genes. These genes have been linked to gastrointestinal functioning and disease, for example, an increased rate of liver injury [29] and biliary cirrhosis [30]. This can be directly (or indirectly) associated with the gastrointestinal ADRs reported by patients. The pathway associated with neurological ADRs consisted of two genes, GABRG1 and GABRG2. These are associated with neuropathic pain [31], hyperalgesia [32], migraines [32,33], and tension headache [31,32], similar to the clinical ADRs reported by our patients.

The limitations of the current study were the small sample size for a GWAS and not enough RNA for transcriptomic assays and gene expression validation by RT-qPCR.

## 4. Material and Methods

We conducted a 12-month prospective treatment study of adults with chronic CD at two large specialized centers, the Hospital das Clínicas and the Institute of Infectology “Emilio Ribas” in Sao Paulo, Brazil. Eligible patients were aged between 18 and 65 years, seropositive, and PCR-positive for *T. cruzi*. Chagas disease clinical form and treatment indication with BZN were defined according to the Brazilian Ministry of Health Consensus statement [34]. Chronic comorbidities were accessed by self-reporting during study interviews and confirmed by reviewing each patient’s medical record. These conditions were diagnosed in agreement with clinical societies’ guidelines. Exclusion criteria were previous BZN use; advanced Chagas cardiomyopathy (New York Heat Association gradation III or IV); megaesophagus grade IV (Rezende’s classification) [35]; advanced megacolon, hepatic, or renal failure; and inability to attend follow-up appointments. Participants received BZN (5 mg/kg/day) for 60 days, at a maximum dose of 300 mg/day. They underwent regular clinical evaluations that consisted of a medical history and physical examination, a biochemistry panel, and serum analysis. Clinical evaluations were performed 30 days before treatment (-T30d), on the day before BZN initiation (T0d), and on days 15 (T15d), 30 (T30d), and 60 (T60d) after starting treatment. Further study visits were conducted at 6 (T6M) and 12 (T12M) months following treatment initiation. Blood samples were obtained during all follow-up appointments.

ADRs were assessed at T15d, T30d, and T60d by specifically asking about dermatological (pruritus, rash, urticaria), gastrointestinal (nausea, vomiting, abdominal pain, reduced appetite), neurological (headache, paresthesia, or dysesthesia), and joint symptoms, as well as by monitoring for laboratory abnormalities. Hepatotoxicity was defined as an increase in aminotransferases greater than three times the upper normality limit. Nephrotoxicity was defined as an increase in creatinine of more than 1.5 times the upper limit and myelotoxicity as a decrease in hemoglobin, leukocytes, or platelets of more than 10% from pretreatment levels. Data obtained were uploaded on the Research Electronic Data Capture (REDCap) database specifically designed for the present study [36].

Patient characteristics were compared between groups using the chi-square or Fisher’s exact test for categorical variables and a Student’s *t*-test for continuous variables. Relative risk and respective 95% confidence intervals were calculated. All statistical tests were two-tailed with an α error of 0.05. Statistical analyses were conducted using Stata (v.13.0).

### 4.1. Genomic and Transcriptomic Sample Extraction

Blood samples were centrifuged, and separate aliquots of plasma and buffy coat/red cells were taken and stored at −20 °C. Genomic DNA was extracted using a QIAamp DNA mini kit (Qiagen, Hilden, Germany), and the concentration was measured using a NanoDrop spectrophotometer (Thermo Fisher Scientific, Waltham, MA, USA). Samples were accepted for genotyping if they presented a 260/280 ratio between 1.7 and 1.8 or a 260/230 ratio between 2.0 and 2.2. In addition, DNA concentration was required to be above 10 ng/μL. DNA integrity was accessed using an agarose 1.5% gel. A total of six samples did not pass the pre-analytic standards, and genotype testing was not performed. Overall, 96 patients were tested using the Affymetrix assay.

Thirty-nine patients were selected for having transcriptome data assayed using samples collected at T0d and T60d. All 13 patients without ADRs (gNADR) were chosen along with 26 individuals, randomly selected from the 73, who presented with ADRs during treatment (gADR). Whole-blood samples collected using PAXgene tubes were extracted using a QIAcube PAXgene blood RNA kit (Qiagen, Hilden, Germany). The DNase I treatment step was performed during the procedure to remove contaminating genomic DNA. RNA integrity (RIN) and concentration were measured using Agilent Bioanalyzer 2100 with a Nanochip 6000 kit (Agilent Technologies, Santa Clara, CA, USA). All samples had RIN and RNA concentrations higher than 4.5 and 5 ng/μL, respectively.

### 4.2. Transcriptomic Processing and Analyses

Samples were subjected to amplicon RNA sequencing using the Ion Proton Platform. This approach has excellent sensitivity to detect messenger RNA (mRNA) at low abundance. RNA (30 ng) was reverse-transcribed and barcoded. Complementary DNA (cDNA) libraries were generated using an Ion AmpliSeq Transcriptome Human Gene Expression Kit (Thermo Fisher Scientific, Waltham, MA, USA) following the manufacturer’s protocol for blood samples [37]. Library concentrations were determined by qPCR using an Ion Library TaqMan Quantitation Kit (Thermo Fisher Scientific, Waltham, MA, USA). Template reactions were carried out using an Ion PI Hi-Q OT2 200 Kit and then loaded onto Ion PI chips v3 using an Ion PI Hi-Q Sequencing 200 Kit (Thermo Fisher Scientific, Waltham, MA, USA). All samples were sequenced using the Ion Proton System, and samples with less than 2 × 10^6^ reads were resequenced. Ion Torrent Suite 5.0 was used for assessing the quality of the libraries and sequencing runs. Base-calling results for each sample were aligned to a standard reference file (hg19_ampliseq_transcriptome_ercc_v1.fasta) on Ion Torrent Suite (v. 5.04). Files were exported to R (v. 3.6.2). Data were normalized and filtered using the edgeR package. For all individual comparisons, we calculated the log fold-change (logFC) and associated *p*-value for each gene. Genes that were up- or downregulated by a logFC of 1.25 or more and with a *p*-value of less than 0.005 were considered significant. Differentially expressed genes were loaded into Ingenuity Pathway Analysis (IPA) (v. 2.4) (Qiagen, Hilden, Germany) to identify over-represented signaling pathways. IPA identifies the key functional and disease associations for given genes using a score value classification.

### 4.3. Genomic Processing and Analyses

Sample DNA (10 ng) was amplified, enzymatically fragmented, dried, and resuspended following the Axiom 2.0 reagent kit protocol (Affymetrix, Santa Clara, CA, USA) [38]. After the hybridization step, samples were processed in GeneTitan equipment (Affymetrix, Santa Clara, CA, USA), where probes identified 842,007 single-nucleotide polymorphisms (SNPs) per individual. In Axiom Suite Analyzes (Affymetrix, Santa Clara, CA, USA), genotype data were normalized, filtered, and exported. SNP cleaning was performed using PLINK (v. 1.9). Based on genetic population study models, a total of 30,316 SNPs did not follow quality control precepts and were removed. SNP imputation was performed on the Michigan Imputation Server platform (https://imputationserver.sph.umich.edu/ (accessed on 13 November 2020)) using the Haplotype Reference Consortium (HRC) reference panel. To protect against association bias due to the population genetic structure, we used the first 4 principal components as covariates in our association models. Polymorphic markers were filtered by Hardy–Weinberg equilibrium using a *p*-value threshold of 1 × 10^−8^. The generated data were analyzed using PLINK, applying the allelic association model to gADR and gNADR groups and correcting for the population structure using the first two principal components. A *p*-value of less than 5 × 10^−8^ was considered genome-wide significant.

## 5. Conclusions

In conclusion, ADRs, especially dermatological reactions occurring within the first 15 days, were the main reason for abandoning BZN treatment. Although the results are preliminary, it was possible to identify a number of SNPs associated with the manifestation of ADRs. In addition, we were able to demonstrate a relationship between the signaling pathways identified and the clinical manifestations in the gADR group, namely dermatological, gastrointestinal, and neurological ADRs. These findings provide new preliminary perspectives for the management of BNZ-related ADRs, since they describe a future research roadmap that could lead to the use of these genetic findings in CD patients’ treatment. For example, they may allow better phenotyping of CD patients before initiation of BZN and stratification of the risk of adverse reactions, allowing greater attention to be given to high-risk patients, potentially facilitating treatment adherence.

## Figures and Tables

**Figure 1 ijms-22-01960-f001:**
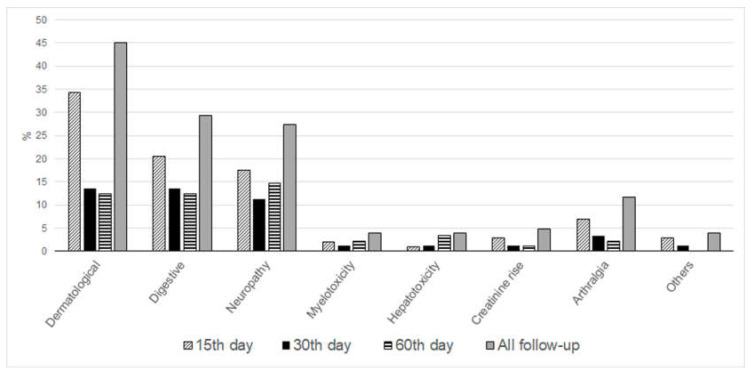
Adverse drug reactions at different follow-up visits: percentage of the whole cohort experiencing each type of adverse drug reaction (ADR), 15 (T15d), 30 (T30d) and 60 days (T60d) after BZN treatment initialization, across the whole follow-up period. ADRs grouped as dermatological, digestive, neurological, and other less common manifestations.

**Figure 2 ijms-22-01960-f002:**
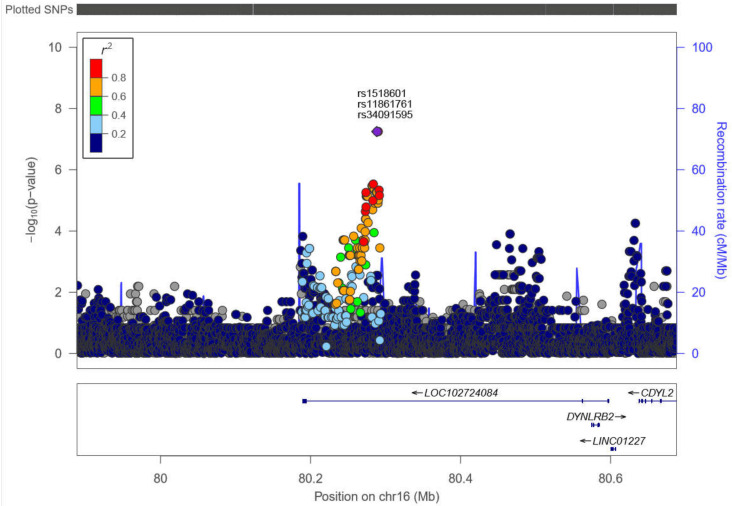
Chromosome 16. SNPs rs1518601, rs11861761, and rs34091595 (*p*-value = 5.65 × 10^−8^) at positions 79888258–80688258 bp of chromosome 16, showing linkage disequilibrium with other adjacent SNPs. The color scale represents *r*^2^ values.

**Figure 3 ijms-22-01960-f003:**
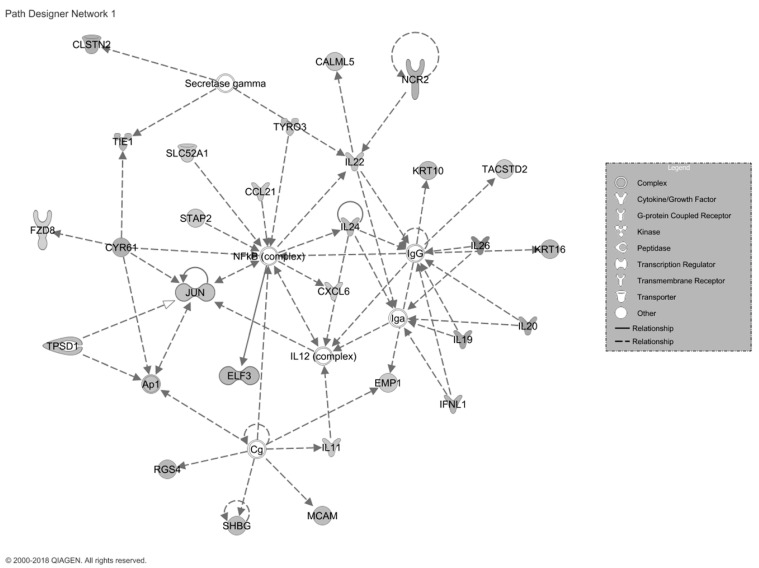
Representation of the dermatological signaling pathway. The 28 filled icons represent genes with increased expression, demonstrating strong activation of the pathway proposed as the main cause of dermatological ADRs.

**Figure 4 ijms-22-01960-f004:**
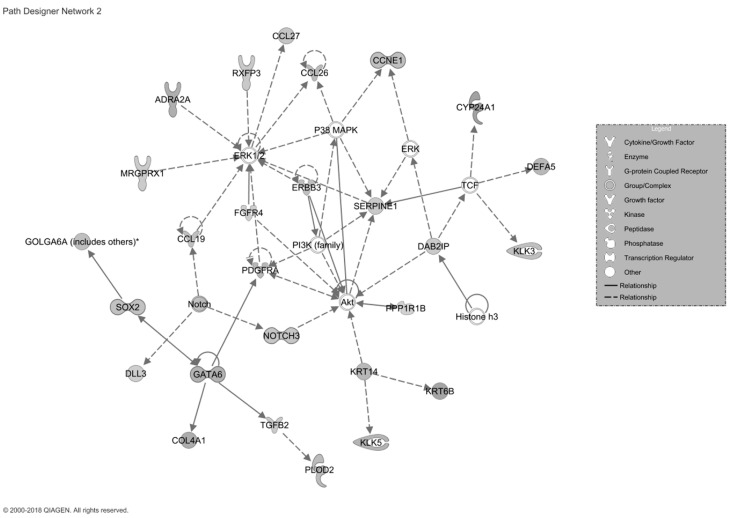
Representation of the gastrointestinal signaling pathway. The 27 filled icons represent increased gene expression, demonstrating activation of the pathway proposed to be associated with the manifestation of gastrointestinal ADRs.

**Figure 5 ijms-22-01960-f005:**
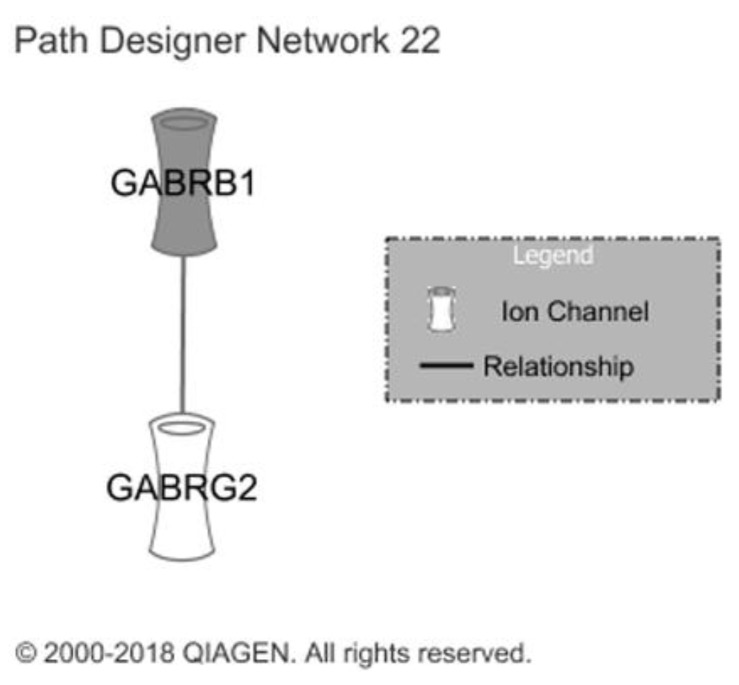
Representation of neurological pathways. The filled icons represent genes with increased expression, indicative of the association with neurological ADR manifestations.

**Table 1 ijms-22-01960-t001:** Demographic and clinical characteristics of Chagas disease patients treated with benznidazole and stratified according to the development (or not) of adverse drug reactions.

Participant Characteristics	Had Adverse Drug Reaction (gADR) *n* = 73 *n* (%) or Median (IQR)	No Adverse Drug Reaction (gNADR) *n* = 29 *n* (%) or Median (IQR)	*p*-Value *
Sex			
Female	35 (47.9)	9 (31.0)	
Male	38 (52.1)	20 (69.0)	0.120
Age, years	53.0 (46–58)	54 (48–59)	0.535 **
Skin color			
Mixed	41 (53.2)	16 (55.2)	
White	22 (30.1)	9 (29)	
Black	9 (12.3)	4 (13.8)	
Indigenous	1 (1.4)	0 (0)	0.932
Literate (yes)	65 (89.1)	27 (93.1)	0.534
Alcohol use (yes)	36 (49.3)	18 (62.1)	0.244
Medication use (yes)	47 (64.4)	21 (72.4)	0.438
Medications by class			
Diuretics	19 (26.0)	4 (13.8)	0.182
ACEi/ARB	21 (28.8)	10 (34.5)	0.571
Beta-blocker	8 (11.0)	2 (6.9)	0.425
Statins	11 (15.1)	5 (17.2)	0.785
PPI	9 (12.3)	6 (20.7)	0.282
Comorbidities			
High blood pressure	32 (43.8)	10 (34.5)	0.387
Diabetes	12 (16.4)	5 (17.2)	0.922
Chronic kidney disease	3 (4.1)	1 (3.4)	0.877
Stroke	1 (1.4)	1 (3.4)	0.495
MI	0 (0)	1 (3.4)	0.111
Clinical forms			
Indeterminate	36 (49.3)	10 (34.5)	
Cardiac	11 (15.1)	13 (44.8)	
Digestive	17 (23.3)	4 (13.8)	
Mixed	9 (12.3)	2 (18.2)	0.016
Completed benznidazole treatment	51 (69.9)	29 (6.9)	0.001

IQR, interquartile range, ACEi, angiotensin converting enzyme inhibitor; ARB, angiotensin receptor blocker; PPI, proton-pump inhibitor; MI, myocardial infarction. * Fischer or chi-square test; ** *t*-test.

**Table 2 ijms-22-01960-t002:** Genome-Wide Association study results: 30 SNPs with the lowest *p*-values.

SNP	CHR	Position	Gene: Consequence	A1	A2	*p*-Value
rs1518601	16	80288258	LOC102724084 Intron variant	A	G	5.652 × 10^−8^
rs11861761	16	80288963	LOC102724084 Intron variant	A	G	5.652 × 10^−8^
rs34091595	16	80289643	LOC102724084 Intron variant	T	C	5.652 × 10^−8^
rs3968927	3	69183404	Leiomodin 3 (LMOD3)/FERM domain containing 4B (FRMD4B) Intergenic variant	T	C	2.945 × 10^−7^
rs6819334	4	165528432	Membrane-associated ring finger 1 (MARCH1)/LOC100133261 Intergenic variant	C	T	3.443 × 10^−7^
rs62255473	3	69182711	Leiomodin 3 (LMOD3)/FERM domain containing 4B (FRMD4B) Intergenic variant	C	T	6.191 × 10^−7^
rs1995665	3	69183140	Leiomodin 3 (LMOD3) (fetal)/FERM domain containing 4B (FRMD4B) Intergenic variant	C	A	6.191 × 10^−7^
rs3097439	15	27871229	Gamma-aminobutyric acid—A receptor, gamma 3 (GABRG3)/Oculocutaneous albinism II (OCA2) Intergenic variant	G	T	6.896 × 10^−7^
rs3098545	15	27871414	Gamma-aminobutyric acid—A receptor, gamma 3 (GABRG3)/Oculocutaneous albinism II (OCA2) Intergenic variant	A	C	6.896 × 10^−7^
rs3097436	15	27872152	Gamma-aminobutyric acid—A receptor, gamma 3 (GABRG3)/Oculocutaneous albinism II (OCA2) Intergenic variant	T	G	6.896 × 10^−7^
rs3097435	15	27872235	Gamma-aminobutyric acid—A receptor, gamma 3 (GABRG3)/Oculocutaneous albinism II (OCA2) Intergenic variant	T	C	6.896 × 10^−7^
rs3098548	15	27873614	Gamma-aminobutyric acid—A receptor, gamma 3 (GABRG3)/Oculocutaneous albinism II (OCA2) Intergenic variant	A	G	6.896 × 10^−7^
rs3097434	15	27873755	Gamma-aminobutyric acid—A receptor, gamma 3 (GABRG3)/Oculocutaneous albinism II (OCA2) Intergenic variant	T	A	6.896 × 10^−7^
rs3098549	15	27874094	Gamma-aminobutyric acid—A receptor, gamma 3 (GABRG3)/Oculocutaneous albinism II (OCA2) Intergenic variant	A	G	6.896 × 10^−7^
rs6026692	20	57661598	SLMO2-ATP5E (SLMO2-ATP5E readthrough)/Mitochondrial ribosomal protein S16 pseudogene 2 (MRPS16P2) Intergenic variant	A	C	6.896 × 10^−7^
rs7563279	2	229499707	LOC101928765 Intron variant	G	A	6.915 × 10^−7^
rs7610262	3	69169466	Leiomodin 3 (LMOD3) Intron variant	C	A	8.807 × 10^−7^
rs9844987	3	69170012	Leiomodin 3 (LMOD3) Intron variant	C	T	9.211 × 10^−7^
rs7963499	12	11917397	Ets variant 6 (ETV6) Intron variant	T	C	9.211 × 10^−7^
rs2619732	5	153877722	LOC100271907/Co-repressor interacting with RBPJ, 1 pseudogene 1 (CIR1P1) Intergenic variant	G	T	1.392 × 10^−6^
rs2612528	4	42721251	LOC101927095/Glutaredoxin, cysteine rich 1 (GRXCR1) Intergenic variant	T	C	1.441 × 10^−6^
rs2569913	2	195964436	LOC101927431/Solute carrier family 39, member 10 (SLC39A10) Intergenic variant	C	T	1.457 × 10^−6^
rs7620430	3	69169602	Leiomodin 3 (LMOD3) Intron variant	A	G	1.871 × 10^−6^
rs2872690	3	69171803	Leiomodin 3 (LMOD3)/ FERM domain containing 4B (FRMD4B) Intergenic variant	C	T	1.949 × 10^−6^
rs6464597	7	144323969	Thiamin pyrophosphokinase 1 (TPK1) Intron variant	C	T	2.367 × 10^−6^
rs1496707	11	91882053	Oxysterol-binding protein-like 9 pseudogene 3 (OSBPL9P3)/Ribosomal protein L7a pseudogene 57 (RPL7AP57) Intergenic variant	G	A	2.367 × 10^−6^
rs9931591	16	80283100	LOC101928248/Dynein, light-chain, roadblock-type 2 (DYNLRB2) Intergenic variant	G	A	2.948 × 10^−6^
rs34880994	16	80281336	LOC101928248/Dynein, light-chain, roadblock-type 2 (DYNLRB2) Intergenic variant	A	G	3.38 × 10^−6^
rs11648009	16	80282101	LOC101928248/Dynein, light-chain, roadblock-type 2 (DYNLRB2) Intergenic variant	C	G	3.38 × 10^−6^
rs1435865	2	229480096	Long intergenic nonprotein coding RNA 1807 (LINC01807)/LOC105373920 Intergenic variant	A	G	3.573 × 10^−6^

SNP, single nucleotide polymorphism; CHR, chromosome.

## Data Availability

The datasets generated during the current study are available in the NCBI repository (https://www.ncbi.nlm.nih.gov/geo/query/acc.cgi?acc=GSE154421 (accessed on 13 November 2020) and https://dataview.ncbi.nlm.nih.gov/object/PRJNA602623?reviewer=qqirgdlauqb50qim1j27npq21f (accessed on 13 November 2020)).

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
