# Peer review of "Pharmacogenomic Profile and Adverse Drug Reactions in a Prospective Therapeutic Cohort of Chagas Disease Patients Treated with Benznidazole"

_ijms, 2021, doi:10.3390/ijms22041960_

Round 1

Reviewer 1 Report

In the present observational comparative study, authors have studied the genetic and transcriptional basis of Benznidazole drug adverse reactions, usually administered to treat Chagas diseased patients. In this study, they enrolled 102 Chagas disease subjects who were on Benznidazole treatment. Further, the genomic analysis was performed by comparing 31 SNPs among the subjects who have shown adverse reactions after getting Benznidazole and the subjects among whom the adverse reactions have not been observed. They suggested that the presence of rs1518601, rs11861761, and rs34091595 SNPs make Chagas patients much more vulnerable to have adverse reactions after having Benznidazole treatment. Although the study findings are interesting and relevant, this can set a guideline before treating Chagas patients with Benznidazole to avoid drug-related adverse reactions. However, there are some shortfalls present in the manuscript, which needs to be addressed properly prior to the acceptance of the manuscript. Specific comments are as follows: 

  1. In the present study, authors have enrolled 102 study subjects for genetic analysis and among them, 73 subjects were with ADRs and 29 subjects without ADRs. Authors need to explain how they have determined sample sizes for genetic analysis. How the level of significance was calculated in GWAS analysis in table 2? Did they measure odds ratio to explain that a mutant allele has significantly over expressed in ADRs group compared to non ADRs? Did they correct their p value with any post-correction test for genotyping data?
  2. Before GWAS analysis, how authors confirmed that entire genotyping data were on Hardy Weinberg equilibrium in order to ensure that allele and genotype frequencies in the enrolled populations remain constant for a particular generation in the absence of other evolutionary influences. 
  3. Authors need to address how they determined genetically defined race/ethnicity in enrolled study subjects on which they generated db SNP based GWAS analysis to exclude the possibilities of genetic drift and ethnic admixtures.  
  4. In table 1 author showcased different demographic characteristics of enrolled study subjects. But they need to mention in the methodology section about the specific criteria they had set to define high blood pressure, diabetes, kidney problem, and stroke among the enrolled study subjects.
  5. They must provide a separate paragraph for the statistical analysis they did for the present work.
  6. In table 1 it’s very unclear how the data have presented (mean/median)? As the sample size is very short, that is why I do recommend providing median with 95% CI level for the data value shown in table 1.

Author Response

Comments and Suggestions for Authors

In the present observational comparative study, authors have studied the genetic and transcriptional basis of Benznidazole drug adverse reactions, usually administered to treat Chagas diseased patients. In this study, they enrolled 102 Chagas disease subjects who were on Benznidazole treatment. Further, the genomic analysis was performed by comparing 31 SNPs among the subjects who have shown adverse reactions after getting Benznidazole and the subjects among whom the adverse reactions have not been observed. They suggested that the presence of rs1518601, rs11861761, and rs34091595 SNPs make Chagas patients much more vulnerable to have adverse reactions after having Benznidazole treatment. Although the study findings are interesting and relevant, this can set a guideline before treating Chagas patients with Benznidazole to avoid drug-related adverse reactions. However, there are some shortfalls present in the manuscript, which needs to be addressed properly prior to the acceptance of the manuscript. Specific comments are as follows: 

1. In the present study, authors have enrolled 102 study subjects for genetic analysis and among them, 73 subjects were with ADRs and 29 subjects without ADRs. Authors need to explain how they have determined sample sizes for genetic analysis. How the level of significance was calculated in GWAS analysis in table 2? Did they measure odds ratio to explain that a mutant allele has significantly over expressed in ADRs group compared to non ADRs? Did they correct their p value with any post-correction test for genotyping data?

This study was originally designed to search for biomarkers of therapeutic response and ADRs with BZN use, by transcriptome assays. However, the samples from this cohort were also used in a large study of our laboratory that assesses the genomic relationship with the clinical outcomes of Chagas disease. Only for the cohort evaluated in this study we had data on ADRs with BZN use, for this reason we were only able to assess the genomic profile of ADR for these 102 patients. Even so, surprisingly, when analyzing the genomic data for these patients, we were able to observe unprecedented and interesting GWAS results, in a preliminary descriptive way.

As described in the manuscript, a p value of less than 5 x 10-8 was considered genome-wide significant. No SNP reached our pre-defined genome-wide significance threshold. Nonetheless, a number of markers showed strong association with ADR occurrence.

As this study and results for GWAS are descriptive and preliminary, we did not assess the odds ratio to explain if a mutant allele has significantly over expressed in ADRs group compared to non ADRs. One of our goals with this publication is to highlight the relation between genomic relationships with the ADRs manifestation by BZN use. With this, opening new possibilities for requesting grants for similar studies with much more patients, who could specifically and accurately assess all pharmacogenomics aspects for BNZ use.

The correction of the P value was not applied with post-tests. 

2.Before GWAS analysis, how authors confirmed that entire genotyping data were on Hardy Weinberg equilibrium in order to ensure that allele and genotype frequencies in the enrolled populations remain constant for a particular generation in the absence of other evolutionary influences. 

We have filtered polymorphic markers by Hardy-Weinberg equilibrium using a p-value threshold of 1E-8. This information was added in the edited Methods section

3. Authors need to address how they determined genetically defined race/ethnicity in enrolled study subjects on which they generated db SNP based GWAS analysis to exclude the possibilities of genetic drift and ethnic admixtures.

This is an important point. Our population is characteristically mixed being mainly represented by three different ancestral genetic components: European, African and Native American. To protect against association bias due to population genetic structure we have used the first 4 principal components as covariates in our association models. This information was added in the revised Methods section

4. In table 1 author showcased different demographic characteristics of enrolled study subjects. But they need to mention in the methodology section about the specific criteria they had set to define high blood pressure, diabetes, kidney problem, and stroke among the enrolled study subjects.

We thank the reviewer for highlight this point. These conditions were accessed by self-reporting and confirmed by accessing the patient's medical record. We rely on the agreement of both reports, as already published [1].

Also, the specific conditions such as diabetes, high blood pressure, Renal problems (which we are replacing the terminology for "Chronic kidney disease") were diagnosed in agreement with the respective clinical society's guidelines national or international, followed by the clinics that respectively assisted the patients for these conditions[2][3] . Therefore, we changed this on the draft line. 192 "...Chronic comorbidities were accessed by self-reporting during study interview and confirmed by reviewing the patient's medical record. These conditions were diagnosed in agreement with clinical societies' guidelines...".

5. They must provide a separate paragraph for the statistical analysis they did for the present work.

We thank for this remark. The statistical issues were briefly described from line 210-213.

6. In table 1 it’s very unclear how the data have presented (mean/median)? As the sample size is very short, that is why I do recommend providing median with 95% CI level for the data value shown in table 1.

We thank the reviewer for highlighting this issue. We have now restructured the table for greater clarity. The only continuous variable is age. As the reviewer has suggested this is summarized by its median and interquartile range.

References

  1. Ye, F.; Moon, D.H.; Carpenter, W.R.; Reeve, B.B.; Usinger, D.S.; Green, R.L.; Spearman, K.; Sheets, N.C.; Pearlstein, K.A.; Lucero, A.R.; et al. Comparison of Patient Report and Medical Records of Comorbidities. JAMA Oncol 2017, 3, 1035–1042, doi:10.1001/jamaoncol.2016.6744.
  2. Sociedade Brasileira de Cardiologia; Sociedade Brasileira de Hipertensão; Sociedade Brasileira de Nefrologia [VI Brazilian Guidelines on Hypertension]. Arq Bras Cardiol 2010, 95, 1–51.
  3. Junior, J.E.R. Doença Renal Crônica: Definição, Epidemiologia e Classificação. J. Bras. Nefrol. 2004, 26, 1–3.

Reviewer 2 Report

These data suggest that part of BZN adverse reactions might be genetically determined and may facilitate patient risk stratification prior to starting BZN treatment. Transcriptomic data revealed three significantly enriched signaling pathways associated with the BZN ADR. This may be a good rationale for earlier marker testing prior to treating Chagas disease patients.

Limitations of the current study are the small sample size for GWAS and insufficient RNA for transcriptomic assays and RT-qPCR gene expression validation. The value of 4.5 RIN is very low, 8.0 RIN is a better standard. However, these are preliminary studies, no less very important, which give rise to more targeted research.

Author Response

Open Review 2

English language and style

( ) Extensive editing of English language and style required
( ) Moderate English changes required
( ) English language and style are fine/minor spell check required
(x) I don't feel qualified to judge about the English language and style

Yes

Can be improved

Must be improved

Not applicable

Does the introduction provide sufficient background and include all relevant references?

(x)

( )

( )

( )

Is the research design appropriate?

(x)

( )

( )

( )

Are the methods adequately described?

(x)

( )

( )

( )

Are the results clearly presented?

(x)

( )

( )

( )

Are the conclusions supported by the results?

(x)

( )

( )

( )

Comments and Suggestions for Authors

These data suggest that part of BZN adverse reactions might be genetically determined and may facilitate patient risk stratification prior to starting BZN treatment. Transcriptomic data revealed three significantly enriched signaling pathways associated with the BZN ADR. This may be a good rationale for earlier marker testing prior to treating Chagas disease patients.

Limitations of the current study are the small sample size for GWAS and insufficient RNA for transcriptomic assays and RT-qPCR gene expression validation. The value of 4.5 RIN is very low, 8.0 RIN is a better standard. However, these are preliminary studies, no less very important, which give rise to more targeted research.

We thank the reviewer for the considerations